# GENERATIVE LANGUAGE-GROUNDED POLICY IN VISION-AND-LANGUAGE NAVIGATION WITH BAYES' RULE

**Shuhei Kurita**
AIP, RIKEN
PRESTO, JST
shuhei.kurita@riken.jp

**Kyunghyun Cho**
Courant Institute, New York University
Center for Data Science, New York University
CIFAR Fellow
kyunghyun.cho@nyu.edu

## ABSTRACT

Vision-and-language navigation (VLN) is a task in which an agent is embodied in a realistic 3D environment and follows an instruction to reach the goal node. While most of the previous studies have built and investigated a discriminative approach, we notice that there are in fact two possible approaches to building such a VLN agent: discriminative *and* generative. In this paper, we design and investigate a generative language-grounded policy which uses a language model to compute the distribution over all possible instructions i.e. all possible sequences of vocabulary tokens given action and the transition history. In experiments, we show that the proposed generative approach outperforms the discriminative approach in the Room-2-Room (R2R) and Room-4-Room (R4R) datasets, especially in the unseen environments. We further show that the combination of the generative and discriminative policies achieves close to the state-of-the art results in the R2R dataset, demonstrating that the generative and discriminative policies capture the different aspects of VLN.

## 1 INTRODUCTION

Vision-and-language navigation (Anderson et al., 2018b) is a task in which a computational model follows an instruction and performs a sequence of actions to reach the final objective. An agent is embodied in a realistic 3D environment, such as that from the Matterport 3D Simulator (Chang et al., 2017) and asked to follow an instruction. The agent observes the surrounding environment and moves around. This embodied agent receives a textual instruction to follow before execution. The success of this task is measured by how accurately and quickly the agent could reach the destination specified in the instruction. VLN is a sequential decision making problem: the embodied agent makes a decision each step considering the current observation, transition history and the initial instruction.

Previous studies address this problem of VLN by building a language grounded policy which computes a distribution over all possible actions given the current state and the language instruction. In this paper, we notice there are two ways to formulate the relationship between the action and instruction. First, the action is assumed to be generated from the instruction, similarly to most of the existing approaches (Anderson et al., 2018b; Ma et al., 2019; Wang et al., 2019; Hu et al., 2019; Huang et al., 2019). This is often called a follower model (Fried et al., 2018). We call it a discriminative approach analogous to logistic regression in binary classification.

On the other hand, the action may be assumed to generate the instruction. In this case, we build a neural network to compute the distribution over all possible instructions given an action and the transition history. With this neural network, we use Bayes' rule to build a language-grounded policy. We call this generative approach, similarly to naïve Bayes in binary classification.

The generative language-grounded policy only considers what is available at each time step and chooses one of the potential actions to generate the instruction. We then apply Bayes' rule to obtain the posterior distribution over actions given the instruction. Despite its similarity to the speaker

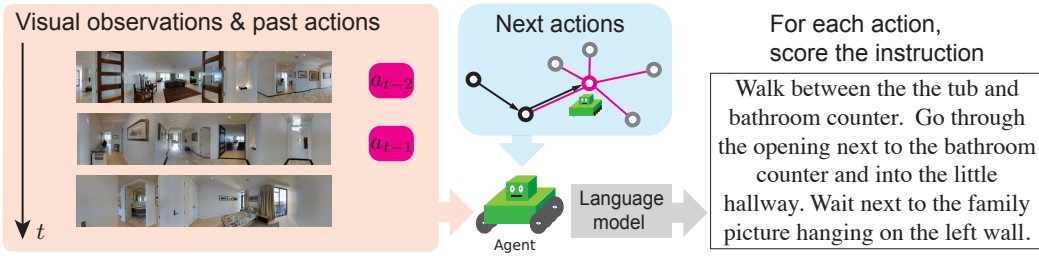

Figure 1: The generative language-grounded policy for vision-and-language navigation.

model of Fried et al. (2018), there is a stark difference that the speaker model of Fried et al. (2018) cannot be used for navigation on its own due to its formulation, while our generative language-grounded policy can be used for it by its own. The speaker model of Fried et al. (2018) takes as input the entire sequence of actions and predicts the entire instruction, which is not the case in ours.

Given these discriminative and generative parameterizations of the language-grounded policy, we hypothesize that the generative parameterization works better than discriminative parameterization does, because the former benefits from richer learning signal arising from scoring the entire instruction rather than predicting a single action. Such rich learning signal arises, because the generative policy must learn to associate all salient features of a language instruction with an intended action, in order to learn the distribution over the language instructions. This is unlike the discriminative policy which may rely on only a minimal subset of salient features of the language instruction in order to model the distribution over a much smaller set of actions. Furthermore, the generative policy enables us to more readily encode our prior about the action distribution when it deemed necessary.

We empirically show that indeed the proposed generative approach outperforms the discriminative approach in both the R2R and R4R datasets, especially in the *unseen environments*. Figure 1 illustrates the proposed generative approach on VLN. Furthermore, we show that the combination of the generative and discriminative policies results in near state-of-the art results in R2R and R4R, demonstrating that they capture two different aspects of VLN. We demonstrate that the proposed generative policy is more interpretable than the conventional discriminative policy, by introducing a token-level prediction entropy as a way to measure the influence of each token in the instruction on the policy's decision. The source code is available at `https://github.com/shuheikurita/glgp`.

## 2 DISCRIMINATIVE AND GENERATIVE PARAMETERIZATIONS OF LANGUAGE-GROUNDED POLICY

Vision-and-language navigation (VLN) is a sequential decision making task, where an agent performs a series of actions based on the initially-given instruction, visual features, and past actions. Given the instruction $X$, past and current observations $s_{:t}$ and past actions $a_{:t-1}$, the agent computes the distribution $p(a_t|X, s_{:t}, a_{:t-1})$ at time $t$. For brevity, we write the current state that consists of the current and past scene observations, and past actions as $h_t = \{s_{:t}, a_{:t-1}\}$, and the next action prediction as $p(a_t|X, h_t)$. The instruction $X$ is a sequence of tokens $X = (w_0, w_1, ..., w_k, ...)$. The relationship between these notations are also presented in Appendix B.

In VLN, the goal is to model $p(a_t|h_t, X)$ so as to maximize the success rate of reaching the goal while faithfully following the instruction $X$. In doing so, there are two approaches: *generative* and *discriminative*, analogous to solving classification with either logistic regression or naive Bayes.

In the *discriminative* approach, we build a neural network to directly estimate $p(a_t|h_t, X)$. This neural network takes as input the current state $h_t$ and the language instruction $X$ and outputs a distribution over the action set. Learning corresponds to

$$\max_{\theta} \sum_{n=1}^{N} \sum_{t=1}^{T_n} \log p(a_t^n|h_t^n, X^n),\qquad(1)$$

where $N$ is the number of training trajectories.

In the *generative* approach, on the other hand, we first rewrite the action distribution as

$$p(a_t|h_t, X) = \frac{p(X|a_t, h_t)p'(a_t|h_t)}{\sum_{a'_t \in \mathcal{A}} p(X|a'_t, h_t)p'(a'_t|h_t)} = \frac{p(X|a_t, h_t)}{\sum_{a'_t \in \mathcal{A}} p(X|a'_t, h_t)}, \qquad (2)$$

assuming $p'(a_t|h_t) = 1/|\mathcal{A}|$, where $\mathcal{A}$ is the action set. This assumption implies that the action is independent of the state without the language instruction, which is a reasonable assumption as the goal is specified using the instruction $X$. $p(X|a_t, h_t) = \Pi_k p(w_k|a_t, h_t, w_{:k-1})$ is a language model conditioned on an action $a_t$ and the current hidden state $h_t$, and outputs the distribution over all possible sequences of vocabulary tokens.

Learning is then equivalent to solving

$$\max_{\theta} \sum_{n=1}^{N} \sum_{t=1}^{T_n} \Big( \log p(X^n|a_t^n, h_t^n) - \log \sum_{a'^n_t \in \mathcal{A}} p(X^n|a'^n_t, h_t^n) \Big). \qquad (3)$$

$\log p(X^n|a_t^n, h_t^n)$ is the language model loss conditioned on the reference action $a_t^n$, while the second term $\log \sum_{a'_t \in \mathcal{A}} p(X^n|a'^n_t, h_t^n)$ penalizes all the actions. Both terms of Eq. 3 are critical for learning the generative language-grounded policy. When we train the model only with the language model term $\log p(X^n|a_t^n, h_t^n)$ of Eq. 3, the resulting neural network may not learn how to distinguish different actions rather than simply focusing on generating the instruction from the state observation.

For navigation, we use the model to capture the probability of the instruction conditioned on each action $a_t \in \mathcal{A}$. The agent takes the action that maximizes the probability of generating the instruction: $\arg\max_{a_t} p(X|a_t, h_t)$. In other words, the language-conditional generative policy has a language model inside and navigates the environment by choosing an action that maximizes the probability of the entire instruction.

## 3 RELATED WORK

While most of previous studies (Anderson et al., 2018b; Ma et al., 2019; Wang et al., 2019; Li et al., 2019; Hao et al., 2020) have relied on the discriminative approach $p(a_t|X, h_t)$, a few of previous studies (Fried et al., 2018; Tan et al., 2019; Ke et al., 2019) have proposed the so-called speaker model which scores the instruction against the entire trajectory. Such speaker models are mainly used for two purposes; (i) data augmentation with automatically generated trajectories (Fried et al., 2018; Tan et al., 2019) and (ii) reranking the complete trajectories in beam decoding (Fried et al., 2018; Tan et al., 2019; Ke et al., 2019). They however have not been used for selecting local actions directly in either training or decoding. To the best of our knowledge, this paper is the first work that propose a standalone generative language-grounded policy for vision-and-language-navigation, that does *not* need the full state-action sequence nor to look ahead into the next state, before taking the action at each step.

Some of the previous studies (Thomason et al., 2019a; Hu et al., 2019) discuss the ablation studies from the multimodal baselines. These studies suggest there are some action biases in the environments. Although it is possible to model these action biases in the action prior of Eq. 2 from the training environment, we choose not to do so in order to avoid overfitting our policy to the training environments. If we know the target environment beforehand, the engineering on the action prior is possibly effective.

Inspired by the the success of the embodied navigation datasets (Wu et al., 2018; Chang et al., 2017; Chen et al., 2019), new experimental settings and navigation tasks in realistic 3D modeling have been proposed, such as dialog-based navigation tasks which include vision-and-dialog navigation (Thomason et al., 2019b), vision-based navigation withlanguage-based assistanc (Nguyen et al., 2019), and HANNA (Nguyen & Daumé III, 2019). Embodied question answering (Das et al., 2018; Wijmans et al., 2019), interactive visual question answering (Gordon et al., 2018) and AL-FRED (Shridhar et al., 2020) for the navigation and object interaction are quite interesting task variants. The proposed generative language-grounded policy is applicable to these tasks where an agent solves a problem by following an instruction or having a conversation with another agent.

## 4 EXPERIMENTAL SETTINGS

### 4.1 DATASETS

We conduct our experiments on the R2R navigation task (Anderson et al., 2018b), which is widely used for evaluating language-grounded navigation models and R4R (Jain et al., 2019), which consists of longer and more complex paths when compared to R2R. R2R contains four splits of data: train, validation-seen, validation-unseen and test-unseen. From the 90 scenes of Matterport 3D modelings (Chang et al., 2017), 61 scenes are pooled together and used as seen environments in both the training and validation-seen sets. Among the remaining scenes, 11 scenes form the validation-unseen set and 18 scenes the test-unseen set. This setup tests the agent's ability to navigate in unseen environments in the test phase. Some of previous studies make use of augmented datasets (Fried et al., 2018; Ma et al., 2019; Tan et al., 2019; Ke et al., 2019) in R2R experiments. We use the same augmented dataset from Fried et al. (2018) which has been used in recent studies (Ma et al., 2019; Ke et al., 2019) for comparison.

R4R was created based on R2R. In R4R, paths are composed of two paths drawn from R2R, implying that each reference path in R4R is not necessarily the shortest path between the starting point and the goal point. R4R is more suitable for evaluating how closely the agent follows a given instruction that corresponds to a long and complex path. R4R consists of train, validation-seen and validation-unseen sets, but does not contain the test-unseen set, unlike R2R. We provide more detailed statistics of R2R and R4R in Appendix C.

### 4.2 NEURAL NETWORK MODELS

We use the network architecture of the speaker from (Fried et al., 2018) to implement generative policies which include a language model $p(X|a_t, h_t)$. We also use the follower network architecture by Fried et al. (2018) for implementing discriminative policies. We follow Fried et al. (2018) and create the embedding of the next action by concatenating the 4-dimensional orientation feature $[\sin \phi; \cos \phi; \sin \theta; \cos \theta]$ and the image feature extracted from a pretrained ResNet (He et al., 2016), where $\phi$ and $\theta$ are the heading and elevation angles, respectively. Both generative and discriminative models use the panoramic view and action embedding, following Fried et al. (2018). The generative policy scores an instruction based on the embedding of each of the next possible actions and the state representation which is also used by the discriminative policy.

**Navigation** In all our experiments, a single agent navigates in an environment only once given a single instruction, for each task, because it is unrealistic to have multiple agents simultaneously navigating in an indoor, hosehold environment. This implies that we do not use beam search nor pre-exploration in unseen environments. See Anderson et al. (2018a) for more discussion on the condition and evaluation of the navigation task.

### 4.3 TRAINING

**R2R** We first train a language model that predict an instruction from the entire trajectory in the same way as Fried et al. (2018) from the dataset. We finetune each policy using imitation learning, where we let the policy navigate the environment and give the action that leads to the shortest path at each time step as supervision, closely following Anderson et al. (2018b). Just like Fried et al. (2018), we start training a policy with both the augmented and original training sets, and then switches to using the original training set alone.

**R4R** we first train a language model for the generative policy from the R4R dataset. Since there are more than 10 times more training instances in R4R than in R2R, we do not augment data. Unlike in R2R, we test both learning strategies; supervised learning and fidelity-oriented learning. In the case of supervised learning, we train both our generative and discriminative policies to maximize the log-probability of the correct action from the training set (Fried et al., 2018). On the other hand, fidelity-oriented learning is a particular instantiation of imitation learning, in which a set of heuristics are used to determine the correct next action based on the proximity of the current state to the reference trajectory at each time step. We describe fidelity-oriented learning in Appendix D.1.

| Model | | | Validation (Seen) | | | | | | | Validation (Unseen) | | | | |
|---|---|---|---|---|---|---|---|---|---|---|---|---|---|---|
| | PL↓ | NE↓ | SR↑ | SPL↑ | CLS↑ | nDTW↑ | SDTW↑ | PL↓ | NE↓ | SR↑ | SPL↑ | CLS↑ | nDTW↑ | SDTW↑ |
| Disc. | 10.69 | 5.40 | 0.519 | 0.482 | 0.619 | 0.588 | 0.445 | 12.88 | 6.52 | 0.380 | 0.335 | 0.488 | 0.458 | 0.304 |
| Disc. +Aug.(A) | 10.60 | 5.15 | 0.525 | 0.489 | 0.633 | 0.596 | 0.445 | 12.05 | 6.22 | 0.431 | 0.392 | 0.528 | 0.496 | 0.356 |
| Gen. | 11.23 | 5.53 | 0.481 | 0.451 | 0.625 | 0.579 | 0.427 | 12.98 | 6.17 | 0.434 | 0.371 | 0.514 | 0.478 | 0.344 |
| Gen. +Aug. (B) | 11.45 | 4.78 | **0.563** | **0.531** | **0.664** | **0.630** | **0.505** | 13.92 | 4.78 | **0.476** | **0.405** | **0.539** | **0.503** | **0.379** |
| Gen.+Disc.(A+B) | 10.18 | 4.67 | 0.568 | 0.540 | 0.680 | 0.640 | 0.510 | 12.06 | 5.42 | 0.489 | 0.437 | 0.570 | 0.533 | 0.403 |
| Gen.+Disc.(A+B)* | 11.30 | 4.58 | 0.575 | 0.541 | 0.678 | 0.636 | 0.509 | 14.65 | 5.19 | 0.518 | 0.439 | 0.564 | 0.515 | 0.397 |

Table 1: Performance of *generative policies* (Gen.) and *discriminative policies* (Disc.) on the R2R dataset. +Aug. represents policies trained with the augmented dataset by Fried et al. (2018). We use (A) Disc.+Aug. and (B) Gen.+Aug. for the combination of the generative and discriminative policies as of (A+B). ∗ represents the use of backtracking of FAST. Bold fonts are used for the best result as a single model in major metrics.

## 4.4 TRAINING DETAILS FOR BOTH R2R AND R4R DATASETS

We use the same neural network architecture with Fried et al. (2018). We use the minibatch-size of 25. We use a single NVIDIA V100 GPU for training. We use the validation-unseen dataset to select hyperparameters.[1]

We use the mixture of supervised learning and imitation learning (Tan et al., 2019; Li et al., 2019) for both the generative and discriminative policies, which are referred as *teacher-forcing* and *student-forcing* (Anderson et al., 2018b). In particular, during training between the reference action $a^{\mathrm{T}}$ and a sampled action $a^{\mathrm{S}}$, we select the next action by

$$a = \delta a^{\mathrm{S}} + (1 - \delta)a^{\mathrm{T}} \tag{4}$$

where $\delta \sim \mathrm{Bernoulli}(\eta)$ following Li et al. (2019). We examine $\eta \in [0, 1/5, 1/3, 1/2, 1]$ using the validation set and choose $\eta = 1/3$.

After both generative and discriminative policies are trained separately, they are combined by

$$\arg\max_{a_t} \left\{ \beta \log p(X|a_t, h_t) + (1 - \beta) \log p_f(a_t|X, h_t) \right\},$$

to jointly navigate in the greedy experimental setting in the R2R dataset. Here $\beta \in [0, 1]$ is a hyperparameter, although our generative model is able to navigate on itself unlike the speaker model by Fried et al. (2018). $\beta$ is determined after the training of both generative and discriminative policies with the same manner. In our experiment, we report the score of $\beta = 0.5$.

FAST (Ke et al., 2019) is a framework of back-tracking to visited nodes. For single-agent back-tracking, FAST(short) adapts a simple heuristic to continue navigation from one of the previously visited nodes. This back-tracking is triggered when the agent choose to visit the same node for the second time. Simple heuristic scoring of the sum of the transition logits is used to choose the returning node. We use this mechanism of back-tracking in the validation and test phase. We use the negative inverse of the logits to determine the node to continue from each time back-tracking is triggered. All movements in back-tracking are counted in the agent trajectory and penalized in the evaluation.

## 4.5 EVALUATION METRICS

We use the following four metrics that have been commonly used to assess a policy in R2R: path length (PL) of the entire trajectory, navigation error (NE), success rate (SR) and success weighted by path length (SPL). Among those evaluation metrics, we consider **SR** and **SPL** as primary ones for R2R because they are derived from the number of successes trials in the navigation. We report CLS (Jain et al., 2019), nDTW and SDTW (Ilharco et al., 2019) in addition to the four metrics above, as these additional metrics are better suited to longer and more complex paths. These three metrics are based on the distance between the policy's trajectory and the reference path. Following Zhu et al. (2020b), we use **CLS** and **SDTW** as primary metrics for R4R. We avoid using SPL for the R4R evaluation because it is not suitable for R4R performance comparisons (Jain et al., 2019). See Appendix E for the detailed description of each metric.

---

[1]For more details of training and evaluations, we closely follow the publicly available code `https://github.com/ronghanghu/speaker_follower` of Fried et al. (2018).

| Model | Validation (Seen) | | | | Validation (Unseen) | | | | Test (Unseen) | | | |
|---|---|---|---|---|---|---|---|---|---|---|---|---|
| | PL↓ | NE↓ | SR↑ | SPL↑ | PL↓ | NE↓ | SR↑ | SPL↑ | PL↓ | NE↓ | SR↑ | SPL↑ |
| Random | 9.58 | 9.45 | 0.16 | - | 9.77 | 9.23 | 0.16 | - | 9.93 | 9.77 | 0.13 | 0.12 |
| Seq2seq | 11.33 | 6.01 | 0.39 | - | 8.39 | 7.81 | 0.22 | - | 8.13 | 7.85 | 0.20 | 0.18 |
| RPA | - | 5.56 | 0.43 | - | - | 7.65 | 0.25 | - | 9.15 | 7.53 | 0.25 | 0.23 |
| Speaker-Follower | - | 3.36 | 0.66 | - | - | 6.62 | 0.35 | - | 14.82 | 6.62 | 0.35 | 0.28 |
| Self-Monitoring | - | - | - | - | - | - | - | - | 18.04 | 5.67 | 0.48 | 0.35 |
| RCM+SIL (train) | 10.65 | 3.53 | 0.75 | 0.67 | 11.46 | 6.09 | 0.50 | 0.42 | 11.97 | 6.12 | 0.43 | 0.38 |
| EnvDrop | 11.0 | 3.99 | 0.62 | 0.59 | 10.70 | 5.22 | 0.52 | 0.48 | 11.66 | 5.23 | 0.51 | **0.47** |
| FAST* | - | - | - | - | 21.17 | 4.97 | 0.56 | 0.43 | 22.08 | 5.14 | **0.54** | 0.41 |
| PRESS | 10.57 | 4.39 | 0.58 | 0.55 | 10.36 | 5.28 | 0.49 | 0.45 | 10.77 | 5.49 | 0.49 | 0.45 |
| Gen.+Disc. Policy | 10.18 | 4.67 | 0.57 | 0.54 | 12.06 | 5.42 | 0.49 | 0.44 | 11.90 | 5.52 | 0.51 | **0.46** |
| Gen.+Disc. Policy* | 11.30 | 4.58 | 0.57 | 0.54 | 14.65 | 5.19 | 0.52 | 0.44 | 14.31 | 5.24 | **0.54** | **0.46** |
| Human | - | - | - | - | - | - | - | - | 11.90 | 1.61 | 0.86 | 0.76 |

Table 2: Comparison of baselines and the proposed policy under single run experimental setting on the R2R dataset. Bold fonts for the first and second best values in SR and SPL. ∗ represents the use of backtracking of FAST.

## 5 RESULTS

We use R2R as the main task to investigate the efficacy of and analyze the behaviour of the proposed language-grounded generative policy and its relative performance against existing approaches, as R2R has been studied more extensively than R4R has. We thus present the result on R2R first, and then finish the section with the result on R4R.

### 5.1 GENERATIVE VS. DISCRIMINATIVE POLICIES

Table 1 shows the performances of the generative language-grounded policy (Generative Policy) and discriminative policy (Discriminative Policy) in the R2R dataset. We show the result with and without data augmentation. All the policies were trained with a stochastic mixture of supervised learning and imitation learning, resulting in better performance than those reported by Fried et al. (2018), even in the case of the discriminative baseline.

The first observation we make is that data augmentation has a bigger effect on the validation-unseen split than on the validation-seen split. This suggests that the main effect of data augmentation is to make a policy more robust to the changes in environments by discouraging the policy from overfitting to environments that are seen during training. This effect is observed with both discriminative and generative policies. However, we consider the discriminative policies are easy to overfit to seen environments in the training time especially without the augmented dataset. In the validation-unseen split, the generative policy always performs better than the discriminative one in both SR and SPL.

Second, when data augmentation was used, the proposed generative policy outperforms the discriminative policy in both validation-seen and validation-unseen splits. This is particularly true with the primary metrics, SR and SPL. The path length (PL) is generally longer with the generative policy, but the difference is within 1 meter on average.

Finally, the best performing approach is the combination of the discriminative and generative policies (both trained with data augmentation). This clearly indicates that these two approaches are capturing two different aspects of visual language navigation. Back-tracking further improves this hybrid policy in terms of SR, although the improvement in SPL is minimal, as back-tracking introduces extra transitions.

In CLS, nDTW and SDTW, the generative policy achieves higher performance than the discriminative policy does, which suggests that the proposed generative policy follows the reference path more closely compared to the discriminative one. We conjecture this is because the generative policy is sensitive to the language instructions by construction.

### 5.2 COMPARISON AGAINST BASELINES

Table 2 lists the performances in the validation-seen, validation-unseen and test-unseen sets in R2R, collected from the public leaderboard and publications. We achieve near state-of-the-art result only

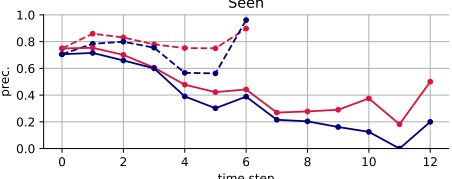 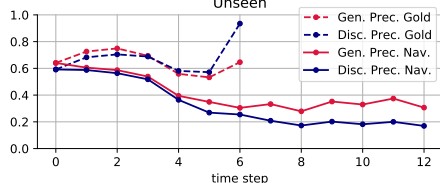

Figure 2: The precision of actions by the generative (red) and discriminative (blue) models on the reference trajectory (dashed lines) and on navigation trajectories (solid lines). The horizontal axis is the time step either on reference trajectory or on navigation trajectory. The vertical axis is the proportion that the agent chooses the shortest path action in each time step.

with the original training set and augmented dataset released by Fried et al. (2018). We compare our approach against the following previous baselines: Random (Anderson et al., 2018b), Seq2Seq (Anderson et al., 2018b), RPA (Wang et al., 2018), Follower (Fried et al., 2018), Self-Monitoring (Ma et al., 2019), RCM (Wang et al., 2019), EnvDrop (Tan et al., 2019), FAST (Ke et al., 2019) and PRESS (Li et al., 2019). They are described in detail in Appendix F. All of them, except for the random agent, follow the discriminative approach, unlike our proposal.

In terms of SR, our model "Gen.+Disc. Policy*" performs comparably to FAST which uses the same neural network by Fried et al. (2018), while our model is better in SPL. In terms of SPL, our model is the second best only next to the EnvDrop model.[2] Our policy however ends up with a better SR than EnvDrop does. Overall, the proposed approach is equivalent to or close to the existing state-of-the-art models in both SR and SPL.

The recently proposed PREVALENT model (Hao et al., 2020) benefits from large scale cross-modal attention-based pretraining. They apply extensive data augmentation to create 6,482K image-text-action triples for pretraining, unlike the other approaches in Table 2. Thanks to this extensive augmentation, they achieve SR of 0.54 and SPL of 0.51. On the other hand, we only use 178K augmented examples from Fried et al. (2018), widely used in previous studies (Ma et al., 2019; Ke et al., 2019), for more direct comparison with previous studies. Although we have nevertheless achieved the comparable SR with an order of magnitude smaller augmented data, we expect our approach would further improve with this more aggressive augmentation strategy in the future.

### 5.3 ACTION PREDICTION ACCURACY

Figure 2 plots the precision of predicted actions over time on the validation-seen and validation-unseen sets in R2R for both the generative policy and the discriminative language-ground policies. We use the discriminative and generative policies notated as A and B in Table 1 for this analysis. When the agents are presented with the gold trajectories, both policies predict actions more accurately than they would with their own trajectories. In real navigation, the action selection error accumulates, and prediction by both policies degrades over time. The generative policy, however, is more tolerant to such accumulated error than the discriminative policy is, achieving a higher precision in later steps. This is especially the case in unseen environments. Additional analyses for the difference of policies are in Appendix G.

### 5.4 TOKEN-WISE PREDICTION ENTROPY

The proposed generative policy allows us to easily inspect how it uses the instruction. A few tokens in an instruction often have significant influence on the agent's decision. For example, if an instruction ends with "...then stop at the kitchen" and the agent is between the kitchen and dinning room, the token "kitchen" decides when the agent predicts the "STOP" action. Since the generative language-grounded policy relies on token-wise scoring of the instruction given each action, we can directly measure how each token in the instruction affects the action prediction. We call this measure token-wise prediction entropy (TENT) and define it as

$$S(w_k) = -\sum_{a_t \in \mathcal{A}} q(a_t, w_k) \log_{|\mathcal{A}|} q(a_t, w_k), \tag{5}$$

---

[2]Our reported SPL is 0.4647 only marginally lower than 0.47 of EnvDrop.

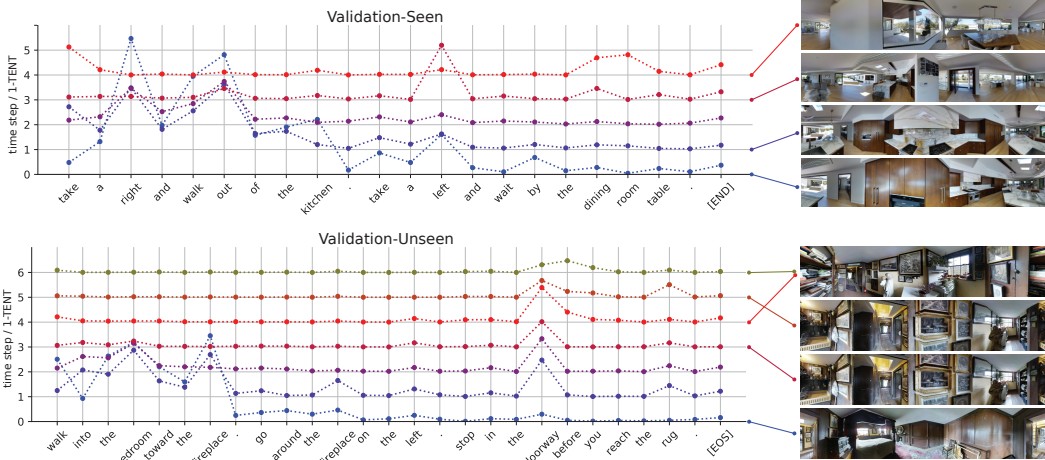

Figure 3: Token-wise prediction entropy (TENT) for two navigation instances from validation-seen (top) and validation-unseen (bottom) sets in R2R. The vertical axis corresponds to the 1-TENT drawn at each time step $t \in \mathbb{N} \cup \{0\}$, as $t + \frac{1}{\Delta}(1 - S(w_k))$, where $\Delta = 0.05$ so that one vertical-tick corresponds to 0.05. We draw multiple lines that correspond to different time steps colored from blue to red and green. We attach the panoramic views for some of the trial time steps.

where $S(w_k) \in [0, 1]$, $\mathcal{A}$ is the action set, and

$$q(a_t, w_k) = \frac{p(w_k | a_t, h_t, w_{:k-1})}{\sum_{a_t \in \mathcal{A}} p(w_k | a_t, h_t, w_{:k-1})}. \tag{6}$$

When some tokens are influential to the action prediction, the entropy of scoring these tokens will be low. Otherwise, when $S(w_k)$ is close to 1, the corresponding token $w_k$ is deemed less influential for the next action prediction. We visualize $1 - S(w_k)$, to which we refer as 1-TENT, to identify highly influential tokens at each time step.

Figure 3 visualizes how actions are related to each token in each time step with two sample navigation trajectories from the validation-seen and validation-unseen splits in R2R. We use the generative policy trained with data augmentation from Table 1. Both trials end successfully within five and seven time steps, and we plot five and seven curves of 1-TENT. In the early stage of the navigation ($t < 3$), initial tokens exhibit large 1-TENT, meaning the change of actions yields a great difference in those token predictions. This tendency is observed in both seen and unseen environments. We conjecture this is a natural strategy learned by the policy to take when there is no navigation history.

In the seen navigation example, the agent is asked to navigate from the kitchen to the dinning room table. In the initial steps, the agent tries to go out from the kitchen, and phrases such as "right" and "walk out" have high 1-TENT. At $t = 3$, the agent is out of the kitchen and needs to turn left at the middle of the large room with high 1-TENT on "left". Finally, the agent finds the dinning table and stops there with the high 1-TENT for the tokens indicating the stop point.

In the unseen navigation instance, the agent is asked to navigate from the hallway, cross the large bedroom and stop outside the carpet. In the trial, the agent first moves toward the goal node based on the keywords "bedroom" and "fireplace". It also exhibits high 1-TENT for "doorway", which is a clue for identifying the goal node. This agent, however, passes the node of the success for the first time at $t = 4$. At $t = 5$, the agent has the high 1-TENT for both "doorway" and "rag" and then goes back to the same place with $t = 4$. Finally, it stops with the high 1-TENT for "before" and the slight 1-TENT for "rag" at $t = 6$. As we have seen here, the agent has different 1-TENT values depending on the context even if it is in the same place.

Although the result of the 1-TENT visualization is similar to the attention maps (Bahdanau et al., 2014; Vaswani et al., 2017), 1-TENT is much more directly related to the action prediction. The attention map represents the internal state of the neural network, while 1-TENT is based on the output of the neural network. This property makes the proposed 1-TENT a powerful tool for investigating and understanding the generative language-grounded policy.

| Model | | Validation (Seen) | | | | | Validation (Unseen) | | | | |
|---|---|---|---|---|---|---|---|---|---|---|---|
| | PL↓ | NE↓ | SR↑ | CLS↑ | nDTW↑ | SDTW↑ | PL↓ | NE↓ | SR↑ | CLS↑ | nDTW↑ | SDTW↑ |
| RCM   fidelity-oriented | 18.8 | 5.4 | 0.526 | 0.553 | - | - | 28.5 | 5.4 | 0.261 | 0.346 | - | - |
| nDTW  fidelity-oriented | - | - | - | - | - | - | - | - | 0.285 | 0.354 | 0.304 | 0.126 |
| BabyWalk  IL+RL | - | - | - | - | - | - | 22.8 | 8.6 | 0.250 | 0.455 | 0.344 | 0.136 |
| BabyWalk  IL+RL+Cur. | - | - | - | - | - | - | 23.8 | 7.9 | 0.296 | 0.478 | 0.381 | **0.181** |
| Disc.  supervised | 20.1 | 7.0 | 0.386 | 0.622 | 0.512 | 0.305 | 20.0 | 9.8 | 0.172 | 0.446 | 0.305 | 0.101 |
| Disc.  fidelity-oriented | 21.1 | 6.6 | 0.449 | 0.644 | 0.530 | 0.360 | 29.2 | 9.2 | 0.211 | 0.385 | 0.282 | 0.116 |
| Gen.  supervised | 19.8 | 8.8 | 0.316 | 0.563 | 0.442 | 0.246 | 19.7 | 9.8 | 0.193 | **0.479** | 0.325 | 0.121 |
| Gen.  fidelity-oriented | 21.0 | 6.9 | 0.448 | 0.629 | 0.517 | 0.349 | 22.8 | 8.7 | 0.255 | 0.471 | 0.348 | 0.162 |

Table 3:   Performance of **gen**erative and **disc**riminative policies on the R4R validation-seen and -unseen splits. We use bold-face to indicate the best models according to CLS and SDTW in the validation-unseen split.

## 5.5   R4R

Table 3 presents the results on R4R with baseline model performance. Similarly to our earlier observation on R2R, the proposed generative policy works as well as or better than the discriminative policy as well as the other baselines, in terms of the primary metrics which are CLS and SDTW in this case especially in the validation-unseen split. The generative policy trained with supervised learning outperforms all the baseline policies in CLS, while the generative policy trained with imitation learning is close to BabyWalk trained with reinforcement learning (IL+RL) and curriculum learning (IL+RL+Cur.)  in SDTW. As both reinforcement learning and curriculum learning could also be applied to our approach, we expect to this gap to completely close in the future.

As we have observed on R2R, without data augmentation, the discriminative policy works as well as or often better than the generative approach does on R4R. the generative policy is however significantly better than the discriminative one in the validation-unseen split, confirming our conjecture that the discriminative policy tends to overfit to environments that were seen during training.

## 6   CONCLUSION

We have investigated two approaches, discriminative and generative, for the vision-and-language navigation task, and presented the generative language-grounded policy which we empirically observed to perform better than the more widely used discriminative approach. We were able to combine the generative and discriminative policies and achieve the (near) state-of-the-art results for the Room-2-Room and Room-4-Room navigation datasets, despite the simplicity of both parameterization and learning relative to the existing baselines. Finally, we have demonstrated that the proposed generative approach is more interpretable than discriminative ones by designing a token-wise prediction entropy.

The proposed generative parameterization, including 1-TENT visualization, is directly applicable to language-grounded reinforcement learning, such as Zhong et al. (2020); Hermann et al. (2017), which should be investigated in the future. The proposed generative parameterization further enables natural integration of large-scale language model pretraining, such as Radford et al. (2018); Brown et al. (2020), for various language-conditioned tasks. It is however important to investigate an efficient way to approximate the posterior distribution in order to cope with a large action set, for instance, by importance sampling and amortized inference, for the proposed generative parameterization to be more broadly applicable, in the future.

ACKNOWLEDGMENTS

SK was supported by ACT-I, JST JPMJPR17U8 and PRESTO, JST JPMJPR20C. KC was supported by NSF Award 1922658 NRT-HDR: FUTURE Foundations, Translation. This work was done when SK visited New York University.

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

## A   EMBODIED AI AND VISION-AND-LANGUAGE NAVIGATION

In the R2R dataset (Anderson et al., 2018b), an agent moves on a graph that was constructed from one of the realistic 3D models of houses and buildings based on Matteport 3D dataset (Chang et al., 2017). At the beginning of each trial, the agent is given textual instruction, is placed at the start node and attempts to reach at the goal node by moving along the edges. At each node of the graph the agent observes the visual features of the surrounding environment and makes a decision to which neighbour node it will move next. When the agent determines that the current node is sufficiently close to the destination node, it outputs "STOP", and the navigation trial ends. The agent is evaluated in terms of the accuracy of their final location and the trajectory length (Anderson et al., 2018b;a).

The difficulties in VLN mainly arise from the diversity of textual instructions. R2R provides multiple instructions for each trajectory. These instructions are created via crowd-sourcing, and their granularity and specificity highly vary (Li et al., 2019). The agent furthermore needs to generalize to unseen environments. Previous studies have reported that models with rich visual and textual features often overfit to the seen environments (Hu et al., 2019).

Recently, VLN becomes the central part of embodied navigation studies (Zhu et al., 2020a; Wang et al., 2020a; Xin et al., 2020; Qi et al., 2020; Wang et al., 2020b; Krantz et al., 2020; Fu et al., 2020; Majumdar et al., 2020). VLN is applicable to real-world robotic navigation even without the preset mapping when it is combined with the external SLAM and autonomous mobile modules (Anderson et al., 2020). Recent embodied AI environments (Savva et al., 2019; Weihs et al., 2020) are also applicable for real-world robotic navigation. Symbol emergence of Taniguchi et al. (2016) might be one clue to close the gap between machine learning-based embodied agents, human interactions, and robotics.

In this paper, we propose the first approach to directly utilize the vision-conditioned language modeling for navigation. Our generative policy applies the vision-conditioned language modeling for scoring the instructions, and therefore our policy implicitly maps the visual information to the language information as visualized in Sec. 5.3. This implicit mapping from visual to language information with the language modeling may contribute to the performance of our generative policy, especially in the navigation of unseen conditions.

## B   RELATIONSHIPS OF NOTATIONS

In the formalism of the generative language-grounded policy, we denote the instruction as $X$, past and current observations as $s_{:t}$ and past actions as $a_{:t-1}$ at time step $t$. Figure 4 illustrates the relationship between these notations. $h_t$ includes the current and past observations and past actions.

## C   DETAILS OF R2R AND R4R DATASETS

R2R has in total 21,567 instructions which are 29 words long on average. The training set has 14,025 instructions, while the validation-seen and validation-unseen datasets have 1,020 and 2,349 instructions respectively. Each trajectory in the R2R dataset has three or four instructions. We use the released augmentation dataset in R2R. This augmentation dataset includes 178.3K trajectories with a single instruction for each. In the R4R dataset, training set, validation seen and validation unseen datasets contains 233k, 1k and 45k instructions respectively.[3] We don't use augmented datasets during the R4R training.

## D   TRAINING DETAILS OF LANGUAGE-GROUNDED POLICIES

### D.1   FIDELITY-BASED TRAINING FOR R4R DATASETS

To let the agent follow the instruction, even if the agent is out of the reference path during student-forcing learning (Anderson et al., 2018b), we introduce a simple heuristics to determine the reference

---

[3]We follow `https://github.com/google-research/google-research/tree/master/r4r` and generate the R4R dataset from R2R.

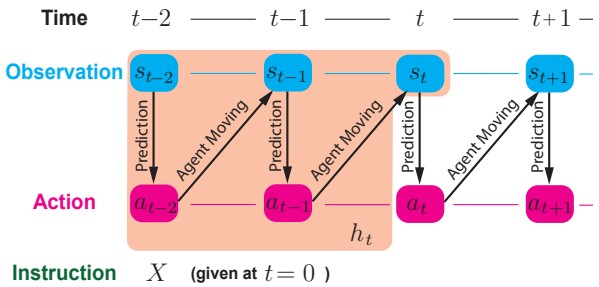

Figure 4: The relation for notations of the instruction $X$, visual scene $s_t$, action $a_t$ and $h_t$ for the VLN agent.

actions. Given the reference path $R = (r_1, ..., r_i, ..., r_{|R|})$ as the sequence of reference places $r_i$ and the current agent trajectory $P_t = (p_1, ..., p_{t'}, ..., p_t)$ as the sequence of visited places $p_{t'}$ at time step $t$, (1) if the current place $p_t$ satisfies $p_t \in R$, the reference action here is the action to follow the reference path including the stop action.[4] (2) if the agent is out of the reference path at time step $t$ and was on the reference path at $t'$, we choose the temporal goal place from the reference path as $\arg \min_r PL(x, R')$ where $R' = (r_i, r_{i+1}, ..., r_{i+t-t'})$. $PL(x, y)$ is the shortest path length between the places $x$ and $y$. Here $r_i$ is the place the agent was last on $R$ at $t'$-th step. $r_i$ is also inferred as the same way with the footnote if $R$ has multiple $x'_t$. The reference action here is the action to lead the agent to the temporal goal place in the shortest path length.

The key idea of this heuristic is that when the agent is out of the reference path, we choose the temporal goal on place from the reference path. However, we disallow the agent to choose the temporal goal place which is far from the instruction and the visiting orders of the reference path.

## E    DETAILS OF EVALUATION METRICS

We use the following four metrics that are commonly used in evaluation for R2R navigation:

**Trajectory Length (TL)** is the length of the agent trajectory in meters.

**Navigation Error (NE)** is the shortest path distance in meters from the point the agent stops to the goal point.

**Success Rate (SR)** is the proportion of successes among all the trials. The task is successful when the agent stops within 3m from the goal point (Anderson et al., 2018b).

**SPL** is short for Success weighted by (normalized inverse) Path Length introduced in Anderson et al. (2018a). SPL is a variation of SR and is penalized by the trajectory length.

We analyze how well the trajectories followed by the proposed approach agree with the instructions using CLS (Jain et al., 2019), nDTW and SDTW (Ilharco et al., 2019). These three metrics are defined as:

**CLS** Coverage weighted by Length Score is the product of the path coverage of the reference path and the length score which penalize the longer or shorter trajectory length than the reference path length.

**nDTW** Normalized Dynamic Time Warping computes the fidelity of the trajectory given the reference path.

**SDTW** Success weighted by normalized Dynamic Time Warping is equal to nDTW for task success cases and otherwise 0.

---

[4]We also consider the case that $R$ has multiple $p_t$ in the R4R training set. When an agent visits the place $p_t$ for the $n$-th time and $R$ include $m$ times of $p_t$ visiting, we assume this is the $m'$-th visiting of $x_t$ on $R$ and $m' = n$ if $n < m$ otherwise $m' = m$. We assume that the agent is at $m'$-th step of the reference path ($p_t = r_{m'}$).

In the R2R dataset, each instruction is based on the shortest path (Anderson et al., 2018b). The trajectory paths are specified only in the instructions and therefore these metrics evaluate how closely the models follow the instructions. Suppose that there are two completely different routes in the navigation: the shortest path with the instruction and a different path that result in a slightly longer path length. When an agent ignores the instruction and reaches the goal on a different route, SPL will be close to 1 because of the similar path length. However, CLS and SDTW are penalized due to the completely different trajectory.

We do not use SPL for the R4R evaluation metric because SPL depends on the shortest path length from the start node to the goal node (Anderson et al., 2018a) and the reference path in R4R is not the shortest path in general. In R4R, the shortest path length from the start node to the goal node can become 0. To consider the fidelity-based path length, we need to re-define a new SPL' based on the reference path length instead of the shortest path length for R4R. However, if we do so, such SPL' is incompatible with SPL reported in the previous paper. Therefore we don't use SPL for R4R performance comparisons. See Jain et al. (2019) for further discussions of SPL on R4R.

## F   DETAILS OF BASELINE MODELS

### F.1   R2R BASELINES

We compare our approach against the following previous baselines in R2R. All of these, except for the random agent, follow the discriminative approach.

**Random**  An agent that moves to one random direction for five steps (Anderson et al., 2018b).

**Seq2Seq**  An LSTM-based sequence-to-sequence model (Anderson et al., 2018b).

**RPA**  Combination of model-free and model-based reinforcement learning with a look-ahead module (Wang et al., 2018).

**Follower**  An agent with panoramic view and trained with data augmentation (Fried et al., 2018).

**Self-Monitoring**  An agent that integrates visual and textual matching trained with progress monitor regularizer (Ma et al., 2019).

**RCM**  An agent that enforces cross-modal grounding of language and vision features (Wang et al., 2019).

**EnvDrop**  An agent trained with combination of imitation learning and reinforcement learning after pretraining using environmental dropout and back translation for environmental data augmentation (Tan et al., 2019).

**FAST**  An agent that exploits the fusion score of the local action selector and the progress monitor. This agent is able to back-track to visited nodes (Ke et al., 2019).

**PRESS**  An agent with the pretrained language encoder of BERT and the capability to incorporate multiple introductions for one trajectory (Li et al., 2019). We compare our model against their model trained with a single instruction.

### F.2   R4R BASELINES

We compare our policies with models that are trained without augmented data if they are available.

**RCM**  An RCM agent that enforces cross-modal grounding of language and vision features reported in Jain et al. (2019).

**nDTW**  An agent that with reinforcement learning with nDTW-based rewards (Ilharco et al., 2019).

**BabyWalk**  An agent that exploits the proposed BABY-STEPs to follow micro-instructions. BABY-STEPs are shorter navigation tasks and trained with other learning regimes such as imitation learning (IL), reinforcement learning (RL) and curriculum reinforcement learning (Cur.).

R4R is a dataset to measure the agent fidelity to the given instruction. Therefore we choose the fidelity-oriented agents in comparison and we develop our policies with supervised learning or fidelity-oriented training.

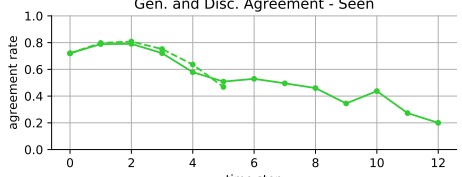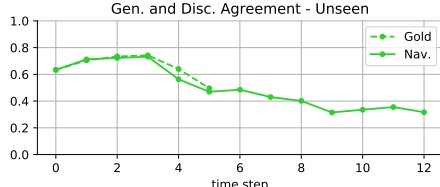

Figure 5: The agreement of actions between the generative and discriminative models on shortest paths (dashed lines) and on navigation trials (solid lines). The horizontal axis corresponds to the time step of trials.

## G    AGREEMENT OF GENERATIVE AND DISCRIMINATIVE POLICIES

We present the agreement rate of action prediction between generative and discriminative policies at the bottom of Figure 6. The agreement drops over time, which implies that these policies behaves differently from each other, capturing different aspects of VLN. The agreement become lower in later time steps and we hence consider the combination of Gen and Disc can work better than either model.

## H    PERFORMANCE COMPARISON ON LONGER TRAJECTORIES

We present further comparisons of the discriminative and generative policies on longer trajectories with the R4R validation unseen set. Figure 6 presents SR, CLS, and SDTW for navigation trials with the total steps in the gold trajectories and the navigation trajectories. In the left pane, the horizontal axis is the number of steps to reach the goal node following the reference trajectory. As seen in Figure 3 of Jain et al. (2019), most of the R4R goal trajectories have 9 to 15 steps. The CLS and SDTW graphs suggest that our generative policy closely follows the goal trajectory compared to the discriminative policy especially when the gold trajectories are long. In the right pane, the horizontal axis is the number of steps when agents stop their navigation in the trials. Although the difference between the generative policy and the discriminative policy is not large in the success rates, the generative policy achieves better in CLS and SDTW with the long navigation trajectories. This suggests that the discriminative policy doesn't follow the given instruction even though it achieves similar success rates to the generative policy.

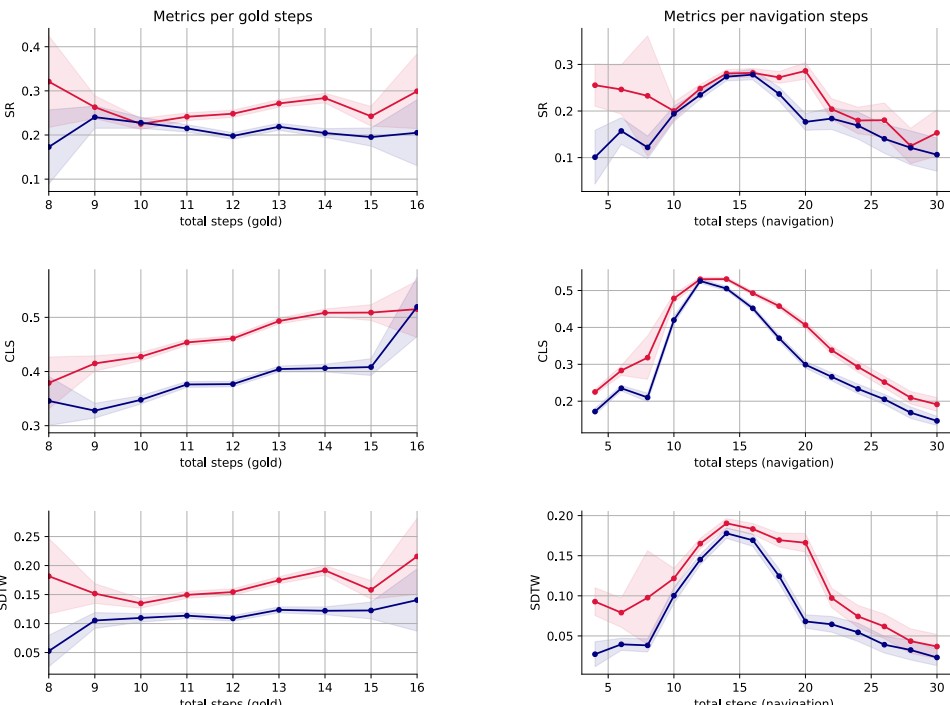

Figure 6: We present SR (top), CLS (middle) and SDTW (bottom) with the confidence interval of $2\sigma$. Crimson for generative policy and cyan for discriminative policy. Both policies are trained with the fidelity-oriented manner.

