# OpenReview forum: "Generative Language-Grounded Policy in Vision-and-Language Navigation with Bayes' Rule"
_ICLR.cc/2021/Conference — ICLR 2021 Poster_

### Official Review · AnonReviewer4 · 2020-10-18
**Reasons of why the proposed method improves the results is not well explained.**

**Rating:** 5
**Confidence:** 4

**Review:**

The paper focuses on learning a navigation policy for a vision-and-language navigation problem. In this problem, the agent are given a language instruction and are asked to follow the instruction to navigation in a simulated 3D room. Unlike baselines which maximize the probability of selecting an action given an instruction, the authors proposed to apply the Bayes rule to maximize the probability of generating the instruction given an action. The authors claim that this gives better generalization in unseen environments.

Pros:
While the idea of using a critic model (to compute the probability of the instruction given the trajectory) for VLN tasks [1,2] is not new, existing works takes the whole trajectory as input to the critic model, yet this paper takes the trajectory up until time t as input. The idea is new.

Cons:
The authors didn't explain well why the proposed generative model is better than the discriminate model. The empirical results shows so, but there aren't enough discussion to explain why.
Experiments are somewhat less satisfying, especially when the results in Table 2 are Gen.+Disc. with backtracking instead of just Gen.+Disc.

Detailed Comments:
It is intuitive that Gen. + Disc. works well than Gen. or Disc. alone. But why Gen. performs better than Disc.? The statement that Gen. works better than Disc. is the main statement and main contribution of this paper. However, there is very little discussion on why this is true methodologically (not empirically). Also, I'd assume that it is extremely hard to learn p(X | a_t, h_t) when t is small (at the beginning of the trajectory). I don't see analysis on how well the instructions are predicted.

Table 1 shows the performance of both Gen. + Disc. and Gen. + Disc. + backtracking. However, In Table 2, the performance of Gen. + Disc. on test (unseen) is not shown. This is less satisfying. Meanwhile, in test (unseen), with backtracking, Gen. + Disc. achieves the same success rate compared with FAST (which also do backtracking), so the performance gain seems small.

The paper misses implementation details. For example, how p(X|a_t, h_t) is modeled is not shown. The authors say "We use the network architecture of the speaker from (Fried et al., 2018) to implement generative policies which include a language model p(X|at, ht)." But the paper is not self-contained. It is better to at least describe the model in appendix.

The paper say "We finetune each policy using imitation learning". However, imitation learning is a very board term, not necessarily mean the student-forcing implied by the paper. It is better to use a more concrete terms.


[1] Reinforced Cross-Modal Matching and Self-Supervised Imitation Learning for Vision-Language Navigation
[2] Vision-Language Navigation with Self-Supervised Auxiliary Reasoning Tasks

---

> ### Author Response · Authors · 2020-11-12
> **Response to AnonReviewer4**
>
> >  It is intuitive that Gen. + Disc. works well than Gen. or Disc. alone. But why Gen. performs better than Disc.? The statement that Gen. works better than Disc. is the main statement and main contribution of this paper. However, there is very little discussion on why this is true methodologically (not empirically).
>
> It is difficult to precisely pinpoint why the proposed generative approach generally works better than the discriminative one. We have at least two conjectures, and believe that they need to more thoroughly and carefully studied and tested in the future. First, we believe a richer learning signal arising from having to predict not only the action by the shortest-path action but the entire language instruction avoids overfitting the policy. Second, we conjecture that the ability to impose the uninformative action prior avoids the generative policy from overfitting to the training environments. Our experimental results indirectly and partly confirm both of these conjectures may be at play, but we believe follow-up investigations are warranted.
>
> > Also, I'd assume that it is extremely hard to learn p(X | a_t, h_t) when t is small (at the beginning of the trajectory). I don't see analysis on how well the instructions are predicted.
>
> In solving VLN tasks, it is only necessary to use p(X | a_t, h_t) to score the given instruction (not the prediction or language generation) and select the accurate action a_t in each time-step. In other words, it is the relative magnitude of p(X|a, h_t) across possible actions a, rather than the absolute ability of generating X given any correct action a’.
>
> In Figure 2, we present the accuracy comparisons between the generative and discriminative policies. The x-axis in Figure 2 corresponds to the number of time steps from the beginning of the transition (t=0) to the end, which we will clarify in the revision. We do not observe any significant difference in the accuracy even when t is small between the generative and discriminative policies.
>
> One interesting observation in this aspect was the large fluctuation in 1-TENT when t was small. This is illustrated in Fig. 3. We do not have any rigorous explanation behind this phenomenon, but believe it is worth future investigation.
>
> > Table 1 shows the performance of both Gen. + Disc. and Gen. + Disc. + backtracking. However, In Table 2, the performance of Gen. + Disc. on test (unseen) is not shown. This is less satisfying.
>
> The results of Gen.+Disc. on test (unseen)  without backtracking is PL: 11.90, NE: 5.52,
> SR: 0.51, SPL: 0.46. Backtracking improves the success rate but SPL is not so altered. We consider this is due to the longer trajectories. We will include this in the revision if needed.
>
> > Meanwhile, in test (unseen), with backtracking, Gen. + Disc. achieves the same success rate compared with FAST (which also do backtracking), so the performance gain seems small.
>
> Our proposed model (Gen. + Disc. with backtracking) performs as best as FAST in the success rate and clearly out-performs FAST in SPL of the recommended R2R metric. We did further analyses on this. In FAST (the FAST-short model in their paper), the backtracking is triggered when the agent tries to revisit some places it has visited. We checked that 49.9% of trajectories of the follower model (by Fried et al. 2018) include revisiting, while 16.8% of the trajectories in our Gen.+Disc.-Policy without backtracking include revisiting in the unseen-validation. This implies that the proposed approach avoids redundant trajectories.
>
> > The paper misses implementation details. For example, how p(X|a_t, h_t) is modeled is not shown. The authors say "We use the network architecture of the speaker from (Fried et al., 2018) to implement generative policies which include a language model p(X|at, ht)." But the paper is not self-contained. It is better to at least describe the model in appendix.
>
> Thanks for your suggestion. We will detail the network architecture and associated hyperparameters in the appendix.
>
> > The paper say "We finetune each policy using imitation learning". However, imitation learning is a very board term, not necessarily mean the student-forcing implied by the paper. It is better to use a more concrete terms.
>
> This is a great point. We will clarify it by explicitly mentioning that we used student-forcing.

---

### Official Review · AnonReviewer1 · 2020-10-22
**Lack of novelty; Mediocre results**

**Rating:** 4
**Confidence:** 4

**Review:**

**Paper Summary**

The paper addresses the problem of vision-and-language navigation (Anderson et al., 2018). The idea of the paper is to use a generative policy where a distribution over all instruction tokens given the previous actions is computed. The agent takes the action that maximizes the probability of the current instruction. The paper reports the results on R2R and R4R datasets.

**Paper Strengths**

- The paper shows that the proposed generative model outperforms the discriminative formulation of the approach.

- The results have been compared with a number of strong baselines.

**Paper Weaknesses**

- The novelty is limited compared to the speaker model of Fried et al. (2018). It is mentioned in the introduction section that Fried et al. use the entire sequence of actions while this paper uses local information at each step. I do not consider this as a large change compared to the previous work.

- In section 4.3, it is mentioned that the method is initially trained using the entire trajectory similar to Fried et al. (2018). This makes it even more similar to the method of Fried et al. (2018) since it does not just use the local information as mentioned in the introduction.

- Figure 3 and its description in Section 5.4 are very confusing. What are different colors in Figure3? Why is the vertical axis labelled timestep? Isn't it supposed to be 1-TENT? The caption does not make it more clear either. It would be good to clarify.

- Two qualitative examples in Figure 3 are not sufficient. Quantitative analysis over the entire dataset should be provided.

- The paper does not respect the page limit. Section 5.3 is about a figure that appears in the appendix.

**Score Justification**

The idea of the paper is not novel. The proposed method does not provide strong results either (it is outperformed by some other methods). Therefore, I cannot justify acceptance for this paper.

**Post-rebuttal comments**

I read the rebuttal and other reviews. I am still not convinced that there is a big difference between this method and speaker-follower of Fried et al. The speaker model cannot be used for navigation by itself but the learned distribution is used to adjust the actions taken by the follower. The only difference between these two methods is that one uses the entire trajectory and the other one uses partial trajectories up to the current time step (pre-trained on the entire trajectory though). Also, as mentioned by another reviewer, some analysis should be provided about why the generative model works better. Due to these issues, I keep my original rating.

---

> ### Author Response · Authors · 2020-11-12
> **Response to AnonReviewer1**
>
> > The novelty is limited compared to the speaker model of Fried et al. (2018). It is mentioned in the introduction section that Fried et al. use the entire sequence of actions while this paper uses local information at each step. I do not consider this as a large change compared to the previous work.
>
> It is regretful the reviewer does not find our proposal a significant departure from the existing approaches, although we believe this may be due to misunderstanding, rather, which we will try to address and clarify in the future revision of the manuscript.
>
> A major innovation in our manuscript is to use Bayes’ rule at each timestep separately to build a generative policy that can be used on its own for vision-and-language navigation. This is unlike the existing approaches that are predominantly discriminative and do not enable the separation between the action prior and instruction-conditioned policy. This is also unlike the speaker model from Fried et al., because the speaker model simply cannot be used for navigation on its own due to its formulation.
>
> Our approach, albeit simple, is principled and works as well as or better than any existing discriminative policy, validating our motivation behind choosing to work in a generative setup. We believe this presents a new direction of investigation in VLN, which is significant.
>
> > In section 4.3, it is mentioned that the method is initially trained using the entire trajectory similar to Fried et al. (2018). This makes it even more similar to the method of Fried et al. (2018) since it does not just use the local information as mentioned in the introduction.
>
> There is no extra information used by our policy compared to what was available to the speaker model by Fried et al. These two policies are simply different from each other.
>
> > What are different colors in Figure3? Why is the vertical axis labelled timestep? Isn't it supposed to be 1-TENT?
>
> In Figure 3, colors correspond to different time steps. We draw one 1-TENT curve for each time step. We use several colors for better readability of multiple curves for multiple time steps.
>
> In the y-axis of Figure 3, we present both time steps and the magnitude of 1-TENT. Here we slide the 1-TENT curves of different time steps to the y-direction by 0.05 of the 1-TENT measurement in order not to overlap curves. Therefore the y-axis of Figure 3 represents both different time-steps and the magnitude of 1-TENT. As you mentioned, we notice the y-label of Figure 3 is somehow misleading and revise it to clarify this.
>
> We present Figure 3 and 1-TENT visualizations in order to examine how the generative policies determine the next actions given the instruction tokens for each time step. Our motivation is that the generative policy has some key-tokens to predict next actions and such key-tokens are changed in time steps. Therefore we choose this y-axis for both the different time-steps and 1-TENT magnitude.
>
> > Two qualitative examples in Figure 3 are not sufficient. Quantitative analysis over the entire dataset should be provided.
>
> In terms of the 1-TENT visualization, we don’t consider that it is possible to provide quantitative analyses over the entire dataset because each 1-TENT visualization is closely correlated to the given instruction and the given instruction differs in each instance.
>
> We believe our reporting of various metrics such as success rate and others serves the purpose of quantitatively understanding the behaviour of the proposed generative approach.
>
> > The paper does not respect the page limit. Section 5.3 is about a figure that appears in the appendix.
>
> It was our mistake. We were in fact referring to Fig. 2 not Fig. 5, in Section 5.3, as was also noticed by R2. We will fix it in the revision.

---

### Official Review · AnonReviewer3 · 2020-10-22
**A solid technical contribution that is concisely presented and analyzed.**

**Rating:** 8
**Confidence:** 3

**Review:**

This paper presents a generative speaker model that selects actions at each timestep that facilitate the generation of the instruction, and demonstrates that a combination of discriminative and generative action prediction models outperform either alone, with the generative model primarily facilitating better generalization to unseen environments.
It would help to have more details in the paper rather than relegated to the appendix, maybe at the expense of the long TENT analysis? I didn't get too much out of that, but would have liked to see more details of the main algorithm in the paper (e.g., details of how the discriminative/generative combo was done).

Improvements:

- Assumption that p(a_t|h_t) is uniform is reasonable, but might be improved by using priors from the training data since there are strong action-conditioned biases in VLN R2R [ https://arxiv.org/abs/1811.00613 Figure 2 ]; some geometric form of this likely holds for the panoramic setting as well (especially for what were formerly "forward" actions -> continuing along headings close to previous heading).

- The (A) and (B) notation in Table 1 was super confusing and it took me a while to figure out it was just an alias for Disc + Aug versus Gen + Aug since that isn't written explicitly anywhere.

- That the combination of discriminative and generative policies gives best results is a major point in the paper, but the combined policy description is relegated to the appendix which feels weird.

Nits:
- typo Introduction Chang et al. is \citet but should be \cite
- Possible typo after Eq (3), "penalizes all the actions" should this be "penalizes all actions except a_t", or is a_t penalized as well (e.g., global optimum is zero)?
- 5.3 typo "Figure 5" -> "Figure 2"
- Typo 5.5 "as good as" -> "as well as"
- Typo 6 "parametrization"

---

> ### Author Response · Authors · 2020-11-12
> **Response to AnonReviewer3**
>
> Thank you so much for your comments and considerations!
>
> > Assumption that p(a_t|h_t) is uniform is reasonable, but might be improved by using priors from the training data since there are strong action-conditioned biases in VLN R2R; some geometric form of this likely holds for the panoramic setting as well (especially for what were formerly "forward" actions -> continuing along headings close to previous heading).
>
> Thank you for your suggestion. The suggested paper addresses the valuable insights for the action prior. There are two aspects to considering this. First, if we assume that the environment distribution is stationary across training and test sets, it will indeed be helpful to use the action prior estimated from the training environments. Second, your suggestion is quite interesting in that if we had access to target environments in the test time or had some indications about target environment distributions, we can estimate the action prior before deploying a trained policy by replacing it with the uninformative action prior we used. In both cases, our generative formulation naturally enables us to easily swap the action prior accordingly.
>
> > The (A) and (B) notation in Table 1 was super confusing and it took me a while to figure out it was just an alias for Disc + Aug versus Gen + Aug since that isn't written explicitly anywhere.
>
> Yes, this is true (we are thankful for your careful reading). We found it difficult to include the detailed instructions of notation (A) and (B) in the current manuscript due to the space limit. In the final revision, we will explicitly specify the relations and aliases of these models.
>
> > That the combination of discriminative and generative policies gives best results is a major point in the paper, but the combined policy description is relegated to the appendix which feels weird.
>
> Thank you for this comment. We will move this detail from the appendix into Section 4 in the revision.
>
> > Possible typo after Eq (3), "penalizes all the actions" should this be "penalizes all actions except a_t", or is a_t penalized as well (e.g., global optimum is zero)?
>
> Thank you for this suggestion. We penalized a_t as well. Therefore, as you mentioned, the training for the a_t is canceled in Eq. 3 while all other actions are penalized.

---

### Official Review · AnonReviewer2 · 2020-11-03
**adds the missing piece in VLN literature**

**Rating:** 8
**Confidence:** 5

**Review:**

This paper introduces the generative modeling of the task of vision and language navigation. At each timestep, for each action, the generation of the instruction is scored. This modeling achieves the-state-of-the-art or competitive results on standard benchmarks Room-to-room (R2R) and Room-for-room (R4R). The proposed generative modeling also allows us to interpret how the model process the instruction. To do so, token-wise prediction entropy (1-TENT) is introduced. The core idea is that if a token is critical for the trajectory, the entropy of scoring the token for all actions will below. Thus by visualizing the 1-TENT for each timestep we can analyze how and when a model fails or what kind of capabilities are missing in the model. Below I list my questions (Q) and suggestions (S):

S1 Introduction 4th paragraph: It was rather hard to understand what's the difference between Fried et. al. and this work. Please try to simplify or add a figure or give examples

S2 Introduction penultimate paragraph: Please clarify the concept of 'richer learning signal'. This paragraph is rather more cryptical compared to the rest of the paper.

S3 Figure2: please add a sentence or two to clarify how this is generated and the punchline of the caption. Also, add the legend for x axis.

S4 Section 5.3: I believe you are referring to Figure 2 not Figure 5 which is in the appendix.

S5 Figure 3: You are trying to achieve a lot with two analyses. Please move one of them to the appendix -- which will give more space to panorama images. Also, the figure is already hard to interpret without the guidance for the level of 1-TENT scores for tokens.

Q1: Section1: The assumption that the action probabilities are uniform given the state makes sense. However, in reality, it's not true. Have you done any experiments on this or any ideas on how to address that?

Q2: Section 4: This might be a lot to ask but is it possible to have results on TouchDown or any other outdoor navigation datasets? Problems are structurally the same however the way people give instructions in outdoor scenes will be different. It would be great to see the generalization of the generative approach to outdoor scenes.

---

> ### Author Response · Authors · 2020-11-12
> **Response to AnonReviewer2**
>
> Thank you so much for your comments and considerations!
>
> > S1 Introduction 4th paragraph: It was rather hard to understand what's the difference between Fried et. al. and this work. Please try to simplify or add a figure or give examples
>
> Thanks for your suggestion. We will think of a way to make it more concise and to-the-point in making the distinction between Fried et al.'s and our approaches. In short, our approach applies Bayes' rule at each step in order to design a generative policy (analogous to Fried et al.'s speaker model) that does not require us to have a full trajectory.
>
>
> > S2 Introduction penultimate paragraph: Please clarify the concept of 'richer learning signal'. This paragraph is rather more cryptical compared to the rest of the paper.
>
> We agree 'richer learning signal' is a vague concept. We will try to clarify it further. In short, we used 'richer learning signal' to refer to the fact that there are more to be predicted when training a generative policy, as it needs to predict the entire instruction at each time step rather than just the action selected from the shortest path.
>
>
> > S3 Figure2: please add a sentence or two to clarify how this is generated and the punchline of the caption. Also, add the legend for x axis.
>
> We will do so. The x-axis is the number of time steps from the beginning of the trajectory (t=0) to the end.
>
>
> > S4 Section 5.3: I believe you are referring to Figure 2 not Figure 5 which is in the appendix.
>
> We sincerely appreciate your careful reading. Section 5.3 mentions Figure 2. We will fix it in the revision.
>
>
> > Q1: Section1: The assumption that the action probabilities are uniform given the state makes sense. However, in reality, it's not true. Have you done any experiments on this or any ideas on how to address that?
>
> Thanks for the suggestion. We agree with you that this is an interesting avenue to pursue. Without knowing what kind of environments a trained policy was put to, which was what we assumed in this paper, we believe it is natural to use the most uninformative prior on the action distribution (p(a|s)). It is however plausible that there are scenarios in which we can take a glimpse at future environments to better estimate this action prior distribution which can then be readily plugged in to our generative policy. We believe this approach and extensions should be studied further in the future.
>
>
> > Q2: Section 4: This might be a lot to ask but is it possible to have results on TouchDown or any other outdoor navigation datasets? Problems are structurally the same however the way people give instructions in outdoor scenes will be different. It would be great to see the generalization of the generative approach to outdoor scenes.
>
> We focused on two in-door tasks (R2R and R4R) while developing and thoroughly investigating this generative policy. Because we have now confirmed the efficacy of the proposed approach, we agree with you that it is the right time to continue on to applying this approach to a more diverse set of scenarios, including outdoor navigation tasks. We hope to pursue this direction in the future.

---

### Author Response · Authors · 2020-11-17
**Manuscript revision**

We have updated the manuscript: revision in the introduction, the caption of Figure 2, adding the Gen.+Disc. Policy results and other minor correction. Please find it for further discussions.

---

### Decision · Program_Chairs · 2021-01-07
**Final Decision**

**Decision:**

Accept (Poster)

**Comment:**

The authors propose to take a token-level generative approach to the task of vision-language navigation (R2R/R4R). The reviewers raise a number of concerns which should be noted in the final version of this work.  The primary concern revolves around generality.  How will this approach generalize to more sophisticated generative and discriminative models?  To what extent is the model relying on the short instruction/action sequences to succeed and would not perform well on longer instructions, longer trajectories, or more abstract language.  Finally, the discussion of the uninformed prior is interesting because while "clean", reviewers note there is no realistic grounded language scenario in which an uninformative prior makes sense.